# Children's Training and Competition in Football: The Coach's View on Family Participation and Healthy Development

Otávio Baggiotto Bettega [1,*], Cesar Vieira Marques Filho [2], Lucas Leonardo [3], João Cláudio Braga Pereira Machado [3], Alcides José Scaglia [4] and Larissa Rafaela Galatti [4]

1.  Fortaleza Esporte Clube, Fortaleza 60590-210, Brazil
2.  Faculty of Physical Education, Catholic University of Brasilia, Brasília 71966-700, Brazil
3.  Faculty of Physical Education and Physiotherapy, Federal University of Amazonas, Manaus 69067-005, Brazil
4.  Faculty of Applied Sciences, State University of Campinas, Limeira 13484-350, Brazil
*   Correspondence: otavio.b.bettega@gmail.com

**Abstract:** The different deployments that involve youth soccer players' development must be discussed and organized by adults, thinking about creating safe, challenging, and stimulating environments for them. Thus, our aim with this study is to investigate variables about family participation in the players' development processes in the context of children's competitions in Brazil. This qualitative–descriptive research was conducted in the under-10 category of the *Nova Liga Gaúcha de Futebol Infantil* (NOLIGAFI) through interviews with twelve coaches and in loco observations. The results showed that all coaches highlighted the importance and role of the family in the healthy development of young players. However, fact-based evidence reveals negative family participation. From this complex scenario, the coaches become fundamental figures for mediating and guiding this process, organizing proposals aimed at educating parents in the sports scenarios to promote an environment of competitive practice in football that is healthier for children.

**Keywords:** youth sports; parents; soccer; coaches; family participation

## 1. Introduction

The different deployments that involve youth soccer players' development must be discussed and organized by adults, thinking about creating safe, challenging, and stimulating environments for them. Emotional support from the family generates an important support base for athletes to face the challenges of the sporting context [1]. However, the excessive pressure and demands found in the family environment act as a demotivator to practice sports [2]. Considering this complex duality, this research seeks to understand how the family participates in the players' development process, with a specific focus on a state football league in Brazil.

The players' development is established by the interaction of several aspects linked to the contexts of participation, the constituted relationships, and the activities developed in the specific process of sports training over time [3–6]. The participation of the family emerges as an important foundation in sustaining the trajectory of the young athlete, both in a sports environment of training and competition, as well as in other social environments [7,8]. In the Brazilian context, where football is strongly expressed in the country's culture, players' exposure and the possibilities of becoming a senior football player further fuel the dreams of young people who start playing the sport [9,10]; in addition, practitioners get involved with the sport as fans, generating an effective social bond [11].

In this trajectory, the family is established as the preponderant axis, making it relevant for parents to develop interpersonal, intrapersonal, and organizational skills for the better management of relationships [12]. Developing such competencies is an essential condition for the children's sports context to be characterized as a healthy environment for soccer practice.

It is important to highlight that the term "health" does not only refer to the absence of diseases, although this is the predominant way in which family members and coaches perceive this concept [13]. From a more robust perspective, the understanding of health established by the World Health Organization (WHO) encompasses a situation of perfect physical, mental, and social well being. The Brazilian Olympic Committee highlights the holistic view of athletes from childhood onwards for the better development of young athletes, including family and coaches as protagonists in this process [14]. Alves and Becker [15] highlight parental involvement as having a direct relationship with the climate established in training and competition environments, interfering with the child's mental state. In this sense, revealing how family members deal with their expectations and demands in these spaces results in direct impacts on the emotional dimensions, as well as their permanence or abandonment of sports practice, and the meaning that they start to attribute to sports throughout their lives, considering the social dimension.

Based on the aforementioned concept of health, we have a broader basis for understanding how it occurs in an athlete's development process. In interviews carried out with Brazilian professional soccer players [16], the results revealed that parents were the main motivators for practicing sports in the initiation stage, but at the same time, they were the main generators of stress. In the systematic review by Coutinho, Mesquita, and Fonseca [17] on parental influence on sports participation, the themes were linked to the type of influence that parents have in the sports environment and the relationship of this influence with the behavioral responses of athletes. The review indicated that the studies present a generalist and superficial characterization of parental influence in sports, but emphasized that positive parental involvement comes from the provision of autonomy and sustained emotional support, associated with higher perceived competence, motivation, fun, and enjoyment for the athlete.

For the healthy development of young athletes, parents need to understand that their behavior in training and competition environments directly interferes with the health and interpersonal relationships of their sons and daughters [18]. It becomes important for parents to select appropriate sporting experiences for their children, balance behaviors between authority and support for autonomy, manage the emotional demands of competition, and establish healthy relationships in the sporting context in an attempt to better understand and help their children to manage demands related to the involvement in the different stages of their development process [12,19].

It is understandable that most of the time, parents do not have the proper understanding of the complexity that the training process involves and how their attitudes and postures directly interfere with the child's health dimension; therefore, it is also the coach's role to help in understanding the purposes of players' development. The Fédération Internationale Football Association (FIFA), through its Grassroots program, highlights some guidelines that coaches can establish with families: (I) at the start of the season or an event, be sure to give parents a quick explanation of the participation, education, and principles of a basic program; (II) be sure to understand and consider the local family culture of the community where the program is taking place, understanding the traditions, beliefs and particular behaviors of families; (III) conduct individual conversations with parents to provide them positive feedback on their child's development and behavior [20].

From this perspective, this study aims to verify perceptions and facts about family participation in players' development processes in the context of state competition in Brazil, the *Nova Liga Gaúcha de Futebol Infantil* (NOLIGAFI), in the under-10 age category. Interviews and observations allow advancing in the discussion of the theme as it seeks data from the perspective of the 12 coaches, mainly responsible for mediating the training process, in addition to the observation of competitions, environments associated with disputes and tensions, with presence and active participation of the crowd in the main state football competition for children in Rio Grande do Sul.

## 2. Materials and Methods

### 2.1. Research Characterization

The research is characterized as qualitative, of the descriptive type. The organization of methodological strategies considers the complexity of the investigated phenomenon, initially tracing parameters for the collection of information, but making the process flexible, identifying new possibilities through interaction with the researched context. To this end, qualitative research uses a naturalist position and an interpretative perspective of human experience, highlighting the observation of social reality [21].

### 2.2. Context and Research Participants

It should be noted that any research on football carried out in Brazil gains unique characteristics, since it is a country with continental dimensions, with more than 213 million inhabitants [22], with football as a "national passion" and a major axis of sports culture. The research's context is focused on a state football league for youth teams in Rio Grande do Sul, a state with more than 280,000 km$^2$ of territorial extension, and a population of over 11.4 million habitants. Only in the age groups covered in this type of competition (7 to 17 years old) are there more than 2 million children and young people [23].

The league in question is the U-10 age category in NOLIGAFI, wich was created in 2012 in the state of Rio Grande do Sul and had 3700 registered players (from 9 to 15 years old) among 21 clubs in 2017. Since then, it has been responsible for competition at the state level for athletes up to 15 years old. The competition for later age groups is now organized by the local Federation. The investigation through interviews was carried out with twelve coaches. Five matches related to the knockout phase of the competition (Quarterfinals, Semifinals, and Final) were also observed. This process led to the researcher traveling to 12 cities located in the central, north, and east regions, covering a total of approximately 2000 km, considering that all interviews and observations were carried out in person. The coaches interviewed were between 23 and 59 years old and had worked as soccer coaches for between 3 months and 36 years of experience. Of the 12 coaches, 7 have a Physical Education degree, 3 are academics, and 2 do not have a superior degree in Physical Education (see Table 1). The project was approved by the Research Ethics Committee of the State University of Campinas (CAAE: 92747918.1.0000.5404).

**Table 1.** Research participants.

| Coach | Age | Years as Coach | Years as Coach in the Club | Educational Background |
|---|---|---|---|---|
| C1 | 36 | 19 | 11 | Post-Graduation *Latu Sensu* |
| C2 | 49 | 24 | 10 | Former Athlete |
| C3 | 28 | 4 | 2 | Incomplete Graduation |
| C4 | 34 | 11 | 8 | Superior Graduation Complete and Federation License |
| C5 | 23 | 4 | 2 | Graduation |
| C6 | 59 | 35 | 24 | Graduation |
| C7 | 34 | 12 | 0.3 | Post-Graduation *Latu Sensu* |
| C8 | 30 | 5 | 5 | Incomplete Superior Graduation and Federation License |
| C9 | 31 | 1 | 1 | High School |
| C10 | 23 | 5 | 2 | Graduation |
| C11 | 27 | 1 | 0.3 | Graduation |
| C12 | 24 | 6 | 2 | Incomplete Graduation |

### 2.3. Instruments of Data Collection

For data collection, semi-structured interviews and observation instruments were used. The characteristic of the semi-structured interview is flexibility regarding the questions

asked of interviewees, allowing the interviewer to explore in more depth specific questions that he or she deems essential based on the theoretical basis used for the investigation [24]. Observation allows the researcher to use the sociocultural context of the observed environment (the socially acquired and shared knowledge available to the participants or members of this environment) to explain the observed patterns of human activity [25].

The interviews were personally conducted by the first author with the coaches in their respective cities, with available places and times, and lasted an average of 1 h 30 min. The central theme addressed was the participation of the family in players' development, and the interview script was composed of five guiding questions aimed at the most comprehensive investigation possible about the coaches' perspective, and these questions may have been broken down into smaller questions or not even asked, since its purpose may have been previously addressed in the coaches' speeches. The observation was carried out in a non-participant way, in which the observer kept a distance from the observed events in order to avoid influencing them [26]. That is, the observation was made in a real environment, but without interventions in the investigated processes [27]. In the observation environment, the researcher was looking for a position in the competitive context to visualize the actions of the fans, the coaches, and also the players. In addition, for the analysis of the observed data, it is relevant to visualize and consider the sociocultural context of Brazilian football [28,29].

Data collection was supported by filling out a field diary that was written at the time and also after the matches [30]. The structure of the field diary was based on Bogdan and Biklen [31] in terms of its descriptive character (in which the concern is to capture an image in words of the place, people, actions, and observed conversations) and reflective (to capture the observer's point of view, ideas, and concerns). The guiding parameters for the construction of the field diary were the same used for the interviews, systematized based on the description of events, report of thoughts and feelings, evaluation of positive and negative facts, and analysis of established meanings [30].

*2.4. Data Analysis*

The research data were collected and finalized in the second half of 2018. After collection, the data remained for three months without being manipulated, establishing a distance from the material and possible undesirable biases [32]. Then, the audio of the interviews was transcribed by the researcher and, after a period of one week, the material was revised to ensure the accuracy of the transcripts [27].

The investigation of the data obtained was based on Yin [33], who proposes an analytical process of qualitative bias composed of five phases: (1) the compilation is aimed at the careful and methodical organization of the original data, allowing a more rigorous analysis of the data; (2) the decomposition establishes the compilation of data in smaller fragments, indicating descriptive labels to each set of decomposed data; (3) the recomposition allows the researcher to question himself and question his or her data, seeking to find different arrangements and combinations arising from the information obtained, which may allow greater visibility and analytical understanding by developing the first phase of analysis, the compilation, which includes the organization and reading of the data, becoming familiar with the contents of the records; (4) the interpretation is carried out with the data that were recomposed in the previous step, in which the researcher mobilizes interpretive skills to create a narrative to present the results; (5) the conclusion finalizes the analytical process, establishing the moment in which a complete understanding of the research is sought. The decomposition and recomposition process took place inductively, in which the thematic groupings were not previously established, but rather that the data led to the emergence of categories [34]. The analysis category of observations (Facts) originated in a deductive way, from an initial categorization [33]. Therefore, the four categories obtained and through which the results were presented are (I) Importance; (II) Role; (III) Facts; (IV) Presence.

To ensure the reliability of the analytical process, which according to Yin [33] is aimed at minimizing errors and biases in a study, two procedures were adopted. The review by the researcher himself [32] consists of each step being revisited and reassessed by the researcher, with a time interval that allows for attenuating the biases derived from their proximity to the data. A period of three months was established between the collection and the beginning of the analysis process, and there was at least a week of interval between the completion of each stage and its review. Another strategy to ensure reliability was the conference by members [32], in which one of the co-authors, who until then had no contact with the data collected, reviewed each of the steps.

To maintain the epistemological coherence of this study, situated as qualitative, with a naturalist position and an interpretative perspective of human experience, quantitative procedures were not used in the data analysis. The research is justified and presents its main findings, and departs from the qualitative vision maintained during all stages of its construction [35,36]. For the initial presentation of the results, the importance and role of the family in the healthy development of young players were classified using positive and negative categories, using two classification strategies: (1) based on interviews with the coaches themselves, the family's behaviors were classified as positive when the coaches listed them as enabling, contributing, helping, attributing security, and encouraging young people to remain engaged in sports, or negative when they evaluated that the family acted in the opposite direction, making the practice of football impossible or difficult or disturbing, attributing insecurity to, and discouraging the practice of football; (2) data from observations systematized in the field diary were also organized into positives and negatives based on the elements indicated in the literature about the healthy development of young players in soccer [17,20,37–41]. As positive elements, there is intervention in competitions with the aim of stimulating pleasure and fun and not just performance, giving positive and constructive feedback, respecting the athlete's autonomy, not seeking to override the coach's guidelines, valuing successes, and being understanding with mistakes. As negative elements, there is the exacerbated demand in relation to sports performance, directive feedback (such as orders to be followed), the exaltation of mistakes, and aggressive language in communication with referees, coaches, and athletes.

From this data organization, the following categories emerged: (1) "The positive aspects of the family participation", highlighting the importance of the family's role and the importance of the family's presence; (2) "The negative aspects of the family participation", with emphasis on the impacts of the family´s presence; and (3) "The Contextual influences on negative and positive family participation". The results based on this categorization are presented below.

## 3. Results

The results showed that all coaches highlighted in interviews the importance and role of the family in the healthy development of young players, and most coaches indicated that family participation in the competition helps more than hinders. These reports took place from a positive perspective regarding players' development. However, when the coaches reported facts related to the family's participation and also from the field diary evidence, the family's participation was negatively evidenced. Figure 1 presents the data and evidence of the positive relations in terms of importance, role, and presence of the family and negative ones through the facts reported by the coaches and observed in a real situation.

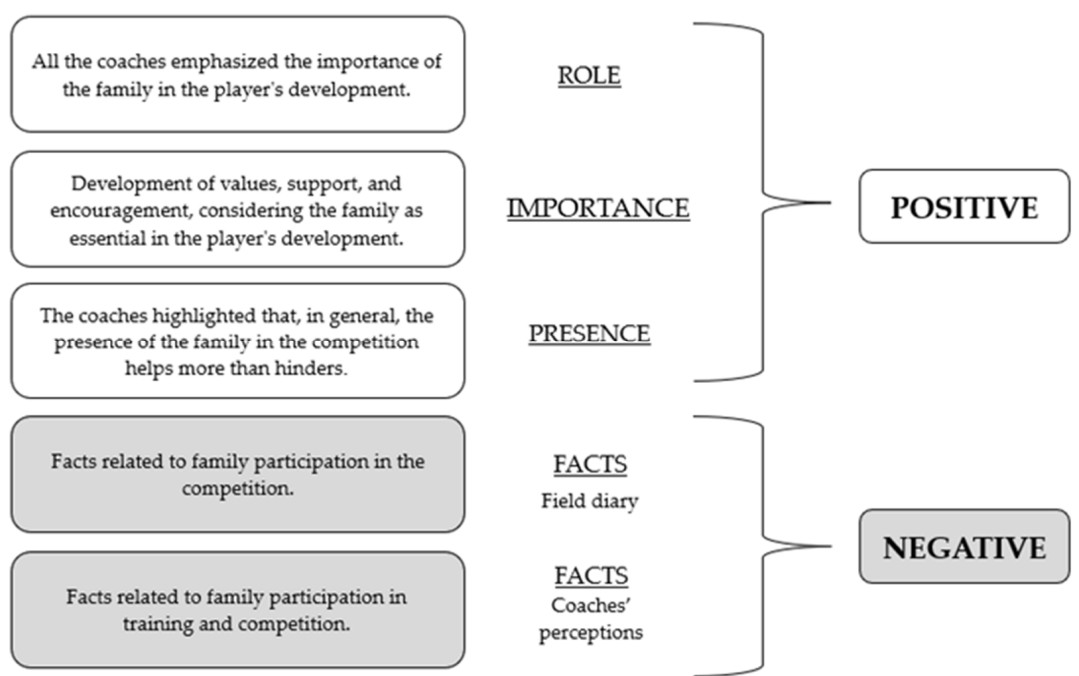

**Figure 1.** Representation of analysis categories and data approach.

*3.1. The Positive Aspects of Family Participation*

3.1.1. The Importance of the Family's Role

In the category of the family's role, the coaches highlighted the development of values, support, and encouragement.

> *It is in the family that we develop values, that we develop the character of anyone, regardless of their age [ . . . ]. (C1)*

> *[ . . . ] I think that family's role is super important for him to be obedient in training, respect his colleagues, and respect the teacher especially, you see cases of great players there who withered because they did not have a structured family base [ . . . ]. (C2)*

Players' development that takes place outside the training and competition environment is highlighted, as well as the stimulus and structure coming from the family to face the challenges and difficulties of the sports environment.

> *The family is important, it has to support, if the student, if the family and the father think that the player will soon be able to follow a professional career, he has to be present [ . . . ]. (C4)*

> *[ . . . ] The family is the base [ . . . ] the family is the main pillar [ . . . ]. (C7)*

> *Man, I would say it's much bigger than the technical and tactical part. I think that when you have a boy whose father doesn't criticize, who encourages and supports, not that incentive of "open it here, open it there", who tries to give guidance, he feels good, and that gives a lot of support, I would say it is essential. (C11)*

3.1.2. The Importance of The Family's Presence

Although coaches agree that the family is essential in training, we realize that in many cases, this importance is justified by the fact that it does not exert negative influences. In many moments, it is indicated that the family contribution to a healthy practice is given by not exceeding demands and adding pressure, especially in a competitive environment.

> *I think that the family has a fundamental role, which is supporting the athlete and not demanding, we have a lot in this category at U-10, and there are many cases of parents who end up interfering, interfering with the development of the boys, instead of helping, they demand the boys, demand performance [ . . . ]. (C5)*

*The family is very important for the player, the family is, but within the limit of each one, they have to encourage the child and not hammer the child that is not even said, he has to use the incentive, encourage, come on. (C6)*

*Encourage, not evaluate, and support. I think this is the role of the family, the dream has to be the child's and not the father's [ . . . ]. (C8)*

*It's very important, I think the family has to encourage it, the family just can't put their child against the teacher, I think the family has to seek staff if they have something to say [ . . . ]. (C12)*

### 3.2. The Negative Aspects of the Family Participation
The Impacts of The Family´s Presence

When the coaches approach in a more objective way the importance of the family in the development of the player, the reported facts involving the participation of the parents in the teaching–training process and in the competition call for attention. The coaches emphasized more negative aspects of their experiences, which involved the drop in performance in training due to pressure, parents' separation, or lack of support and care. There were also reported cases of fights in the crowd, charges for errors, and some children leaving training and games crying. There were still situations where parents wanted to take on the role of coach. We present below just one of several reports from each coach that represent the negative impacts of these types of conduct on the part of the children's parents and relatives.

*There was a boy who cried [ . . . ] he cried when his father went to see the game, he picked up the ball and boom, you looked and he was there, what was it, man? Nothing. Then, in this age category, there is that situation that you can take and put, in a situation of this game, the game started, and I don't think it took 2 min and we took him out, he was crying, the father left, walked around the field, "Come away, Go away, you won't play anymore". Don't worry, he's about to come back in. "No, no, no, you don't know anything." Then the boy left and we had no further contact [ . . . ]. (C1)*

*[ . . . ] we were losing the game 1-0 and the parents started to curse each other and when I saw it, we were facing like this, I saw our parents fighting with the parents of another team, and the children looking, and I went to the judge and said to the person in charge of the other team, who was training the team, I said, look, for the game, let's go to the other side and I'll talk to them, if the parents don't separate, if they don't stop, I'll call the game over, we're going to lose the game, but we're not going to see that [ . . . ]. (C2)*

*[ . . . ] the boy is good and the performance is low because of the father or mother putting pressure on, who wants to develop a professional, who wants to be a professional [ . . . ]. (C3)*

*[ . . . ] this year there was a case here, where a boy made a mistake, and the father from outside charged, the boy left the field, took off his shirt, and left, and then after a while, he returned to training right, the father then asked to leave him out of competitions a little bit [ . . . ]. (C4)*

*[ . . . ] we went to a competition, which was football of 7 too and it was very similar, with the screen close to the field, the parents are there, and as the performance went on, and the opponent's crowd started to scream, it started to make noise. So, this left the kids frustrated and they started to miss some moves [ . . . ] and the children started to make mistakes and the parents of our children started to shout from outside to demand the boys and then you could see that the boys started to look at the father, they had a scared face because they didn't know what to do [ . . . ]. (C5)*

*[ . . . ] some parents want to come to the locker room too to talk to the children, right, and we don't let them, half an hour beforehand, we hold them there at the gate, right, and the children come in, and they also want to come in, no, I'll tie the boot, but no that's it to tie the boot, they want to come to give the tips, for the boy to get a little confused [ . . . ]. (C6)*

*[ . . . ] it happened at the end of the week now, even then I won't tell you that it was because of that, I think it contributed, there was a boy, my goalkeeper, before a more important game, the most competitive of their championship. Yeah, he was warming up like this with the goalkeeper coach, and his father came behind the goalkeeper, I don't even know if his father said anything, I don't know if he was giving a message, but I think it was just that one, because the parents were all on one side, just the fact that he left and approached, I think that there triggered even more, well, the boy started to cry, right man [ . . . ]. (C7)*

*[ . . . ] we have a boy, today he is eleven years old, but he has been since he was eight, the boy was very focused, he is very focused, the boy is dedicated, dedicated, dedicated, but when he makes a mistake, he then starts crying, okay, and then we went to understand why this was happening, the mother makes each game as if it were an event, a world event, she gives a dimension to the game, sometimes a game, in a futsal championship, she makes it, in short, she ends up putting pressure on an unnecessary game, and the boy, when making the mistake, he, he, reacts, he gets disappointed with himself [ . . . ]. (C8)*

*[ . . . ] I had this boy who missed a pass, this defender of mine, and we took the goal, the game was zero to zero against the Juventude team, and they were putting pressure on us, and we held back so as not to take the goal and we took the goal, and the father started to guilt his son, "raise your head, hit the ball", totally the opposite of what the coach asks, and then the boy sank, started cursing the others, right? [ . . . ]. (C9)*

*We had a father, can I give an example? [ . . . ] He went to fatten the kid four kilos, five kilos and took him to the nutritionist. The boy doesn't play anything anymore, he can't, he doesn't have agility, he doesn't have one-against-one, he doesn't change direction, he has nothing. And then it was the father's behavior, he didn't even come to talk to us, "hey, what do you think about this and that?". And then later, when he came to talk to us there, "What do you think about the boy?". "Wow, he's slower and I don't know what, I put on four, five kilos, I took him to the nutritionist, I thought he needed to gain strength". (C10)*

*[ . . . ] a boy who declined a lot in training, like that, and we were putting pressure on him, and we ended up realizing that there was something more, we went to talk, the father was leaving the boy a little aside, he was going to the training session without eating. (C11)*

*We had a situation where the parents split up and the boy was with his mother and he wanted to go to his father, he ran away and didn't tell anyone, he disappeared from training, the family looked for us and the boy didn't miss any training [ . . . ]. (C12)*

### 3.3. The Contextual Influences on Negative and Positive Family Participation

The family's behavior in the competition environment varied based on the circumstances of each match, as well as based on the place where the game took place. In moments of greater dispute and closeness of performance and result, parents exerted more pressure on their children and on the refereeing, and also complained about the coaches. In certain situations, some parents exceeded the demands and ended up directly and negatively influencing their children's participation, as shown in Table 2.

Coaches reported that family participation helps more than it hinders in the competitive environment, as long as it occurs in a balanced way. Only one coach reported that family participation in the competitive context is more of a hindrance than a help.

**Table 2.** Facts related to the family's participation in the competition.

| Facts Related to Family Participation in Competition | Phase |
|---|---|
| In that match, the parents of both teams constantly cursed the referee and charged the children incisively. One parent shouted from the stands, "what a shitty boy", when his son failed to reach a ball and then shouted again, "no use crying now", after his son missed a penalty kick. | Quarterfinals |
| The behavior of the parents in that match was very good; the two crowds shared the stands and tried to cheer and encourage the players, establishing relationships in a harmonious and relaxed way. | Semifinal |
| The game took place under heavy rain, and some parents sought shelter in places further away from the field. The parent's behavior was more restrained in relation to the complaints with the refereeing team, but in the final part of the game, the parents put pressure on the boys and on the referee. | Semifinal |
| In that game, some parents made incisive complaints about the refereeing team, crediting the team's defeat to the referee's fault. A mother of one of the players called the coach of her son's team "dumb", and one of the opposing team's fathers set off fireworks at the start of the game and after his son's team's second goal. | Final |
| The crowd's behavior in that match was somewhat calm; the superiority of one of the teams and the consequent imbalance in the confrontation ended up softening the reactions of the family members in the crowd. | Final |

## 4. Discussion

The results indicate, from the coaches' perspective, that parents' participation in the soccer player's development is essential. The coaches emphasize the role of the family in supporting the practice and in the development of values and social determinants of health, corroborating with the research carried out on parents in soccer schools in different Brazilian cities [42,43]. This scenario highlights an understanding already incorporated into common sense, which points to the idea of the importance of the family in the development of athletes in a healthy way. According to Knight et al. [44], in research realized with seventy parents involved in the sports environment, it was evidenced that parents' behavior is important in emotional support and in the promotion of opportunities. Parental monitoring and emotional support, regarding the dimension of mental health, are associated with increased self-esteem, fun, and motivation of young people, and the motivational climate created by parents helps in dealing with anxiety and self-regulation [45]. Other elements linked to social determinants of health, such as healthy eating habits for young people, are also related to parental influence [46].

Most studies on this subject use quantitative questionnaires, intended for family members or athletes themselves, to collect and analyze a high volume of data and generalize the results [38,42,47]. The qualitative bias adopted in this study, through semi-structured interviews, allows for a deeper understanding of the points of view, perceptions, and meanings indicated by the subjects [48], adopting the coaches' perspective. In addition to a superficial and abstract layer in which they indicate the importance and positive influence of the family, when delving into these issues from more objective and factual elements, the negative influences of the family begin to predominate in the coaches' discourse. In the same vein, Ross, Mallett, and Parkes [49] highlighted, in their research on youth sports in Australia, negative results regarding the interaction of parents in the perception of coaches and administrators, as well as in the communication between parents and their children.

In the study by Kaye, Frith, and Vosloo [39], the results showed that the parents' goals regarding their children's performance influenced their emotional state, usually occurring

negatively. It is possible to perceive the relationship of these findings with the reports of the coaches interviewed here, who mention situations in which the athletes cry and abandon the practice due to the pressure and demands exerted by their parents. Still, in the categories of facts, the results obtained by the field diary are also in line with this panorama.

When entering the sphere of negative impacts on the emotional state of athletes, Dunn and collaborators [41] discuss elements of mental health by highlighting that the greater the financial investment made by parents, the greater the feeling of pressure perceived by the children, as well as the lower the satisfaction and pleasure of practice. In another research carried out on young people between 10 and 16 years old participating in team sports in the Extremadura region of Spain, data also indicate that pressure from parents is associated with less satisfaction and pleasure in the practice [40]. Torregrosa et al. [37], with 893 players with an average age of 15 years, highlighted that the greater the active participation of parents in the context of sports practice, the more they also establish directive behaviors, which have negative consequences on the participation of young players.

This directness generates many moments of shock between the referrals given by family members and coaches. The results showed cases in which parents claim that the coaches should be responsible for their children's ways of playing, including attempts to enter the changing rooms moments before the game to carry out such actions. The research by Pulido, Borrás, and Ponseti [47] on parents of soccer players, carried out using questionnaires, highlights that these parents are interested in maintaining a good relationship with their children, with actions of support and respect in moments of games, but need to improve relations with coaches by narrowing communication channels and not trying to superimpose coaches' orientation and behaviors in a directive way.

In the Australian scenario, research by Elliott and Drummond [38], with 34 parents and 52 children between 12 and 13 years of age in soccer, highlighted the participation of parents in situations after the competition, indicating that interrogative behavior on the part of parents should not only emphasize the performance in the competition but consider the child's involvement in training, games, and other living environments, thus stimulating pleasure and enthusiasm for sports practice. In another study by the respective authors, their gaze turns to the sociocultural context, emphasizing that the behavior of parents, even if Australian grassroots football is guided by some guidelines of conduct from the Australian Sports Commission, expresses influences of sociocultural conditions, often presenting inappropriate attitudes towards grassroots football [50].

In the United States, Goldstein and Iso-Ahola [51] evaluated the behavior of 340 parents of young male and female players (between 8 and 16 years old), and the results highlighted that, even if parents have the best intentions for their children in competition, sometimes they end up losing control, and guidance on parental participation can contribute to a more controlled and positive behavior, enhancing social determinants. From this perspective, the importance of the presence of parents in training and competition environments is highlighted, but it emphasizes that positive social influences depend on appropriate behavior. Therefore, it becomes relevant to develop programs and sports competitions that encourage parents' good behavior. Thus, one can see the importance of an educational process aimed at parents that could contribute to the healthy practice of their children.

In the base categories of a football club in Spain, a program called "Entrenando a familias" (Training families), which is supported by three pillars (cognitive, emotional, and aspects related to guidelines and actions), was developed and evaluated, and showed positive results in parents' behavior, mainly in understanding about the development of children, satisfaction with sports results, as well as a greater focus on the process than on the results and greater experimentation of positive feelings [52].

In the North American continent and also in other countries, we can visualize a series of programs and proposals that aim at the positive development of young people through sports [20,52–55], seeking to create positive interactions, appropriate relationships, and safe

environments, facilitating the participation of young people, and raising awareness of the participation of parents, referees, and coaches.

In this context, coaches represent an important figure in the organization and management of sports practices, obtaining not only a specific role in guiding young soccer players but in managing interpersonal relationships between the different characters in the sports environment, seeking joint work with directors, coordinators, and parents [56]. That is, coaches, in addition to understanding and acting from the specificity of training and competition, must guide parents' participation, thus creating solid bases for the positive development of young players.

The increase in health problems linked to the increase in cases of anxiety, depression, stress, and burnout has drawn attention worldwide [57]. Sports, and especially football in the Brazilian context, have great potential to combat these issues. However, parental participation, crucial to the healthy development of practitioners, can become harmful to their mental health when performed in a negative way, without respecting good coexistence behaviors, and without expressing support and understanding for the challenges and difficulties faced in the practice of sports, regarding its technical–tactical, psychological, and social elements [58].

## 5. Conclusions

The sports phenomenon contemplates a complex web of relationships, which in a proximal perspective is established between players, coaches, and coaching staff, but which expands from this specific context and permeates other environments of coexistence. In the case of the development of the soccer player, the family is configured as a core of significant influence, both in situations that occur at home and on established occasions in sports practice. When it comes to the sports scenario, parents, who are important encouragers and motivators of their children, most of the time do not understand the real objectives of participating in youth sports and end up expressing certain behaviors that hinder the positive development of young players, in a way that negatively interferes with social and mental determinants of health.

The parent's participation in the development of players is extremely important, as reported by all coaches interviewed, but the facts show that their behavior in different scenarios (training, competition, and others) can hinder the participation of young players. Thus, coaches become fundamental figures for mediating and guiding this process; considering the importance of the family's presence in the players' development, they need to organize proposals aimed at educating parents in the sports scenarios to promote an environment of competitive practice in football that is healthier for children.

**Author Contributions:** Conceptualization, O.B.B. and L.R.G.; methodology, O.B.B.; Validation, A.J.S. and L.R.G.; formal analysis, O.B.B.; investigation, O.B.B.; data curation, C.V.M.F.; writing—original draft, O.B.B. and C.V.M.F.; writing—review and editing, C.V.M.F., L.L., J.C.B.P.M. and A.J.S.; supervision, L.R.G. All authors have read and agreed to the published version of the manuscript.

**Funding:** This research was funded by Conselho Nacional de Desenvolvimento Científico e Tecnológico, CNPq (BRAZIL), grant number 141554/2016-9.

**Institutional Review Board Statement:** This study was conducted in accordance with the Declaration of Helsinki, and approved by the Ethics Committee of Universidade Estadual de Campinas (protocol code CAAE: 92747918.1.0000.5404 approved on 08/29/2018).

**Informed Consent Statement:** Informed consent was obtained from all subjects involved in this study. Written informed consent has been obtained from the patient(s) to publish this paper.

**Conflicts of Interest:** The authors declare no conflict of interest. The funders had no role in the design of this study; in the collection, analyses, or interpretation of data; in the writing of the manuscript; or in the decision to publish the results.

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
