# Peer review of "Children’s Training and Competition in Football: The Coach’s View on Family Participation and Healthy Development"

_sustainability, doi:10.3390/su15032275_

Round 1

Reviewer 1 Report

I congratulate the authors for the relevance of the topic addressed in this study. Beyond performance, the contribution of coaches to the health of the athletes is aos paramount importance and nowadays is being widely discussed. However there are some issues that need to be addressed for the paper to be considered for publication.

1. The data from this study is mainly from verbal reports and from behavior observation. These qualitative methods constitute the data, they are not facts nor perceptions, they are the data obtained from the methods used in this study. The article should be coherent with its own approach all over the paper.

2. Studies about social constraints of Brazilian footballers are available in the literature and could be used to highlight the relevance of the topic as well as the methods used. Please see the publications from Luiz Uehara and colleagues.

3. on pg 3 the authors used the curious expression  "socially constructed reality". However constructivism and realism are opposing epistemological conceptions. Given that the authors aim to capture reality as it is observed, I suggest to simply remove the word 'construction'.

4. The instruments for data collection, i.e, the interviews and the observation protocols, must be described in detail. As they are, they are not reproducible due to the lack of information. Moreover, information about their reliability should be presented.

5. Linked with 4, the data analysis needs to be expanded. What was the process of decomposition? what was its reliability?  how were the encodings performed? was it guided by a theoretical framework, which guided the search for  conjunctions and meanings?

6. The categories of analysis are judged as positive o negative instead of describing the role of such categories for performance and health (and thus it would be their utility which indicate if they were positive or negative). So how was the judgement of positive or negative achieved? Simply based on the believes of the researchers?

7. The presentation of results needs to be fully re-organized. Instead of presenting sentences from the interviewees, which are not related to the text, all the sentences need to be commented and interpreted by text. In fact, results and discussion should go together in this type of analysis. As they are, the results are simply a place for dropping sentences. What exactly is the result that each sentence as datum is contributing to? how can it be interpreted in relation with the existing scientific literature?

8. About the observation table, there is no description but judgements. Instead of describing the steam of behaviors (this is how observation needs to be described), or counting frequencies of such behaviors (poorer than the previous method, but reproducible), the authors simply present judgments about what they observed: "constantly cursed" what is constantly? 'very good behavior'; 'some parents'; 'behavior was more restrained', 'somewhat calm'... what exactly these judgements mean?

9. At the end of the results section, the authors talk about concepts that were never described before 'some attitudes carried out by parents' (how did the attitudes where evaluated?) and also how does such "situations generate elements of pressure, stress, anxiety, and create an environment that is harmful to the mental health and psychological conditions of children" how do the authors know anything about these, and what does these variables mean for the authors? In fact, what is an healthy environment for the authors?

In conclusion, the authors need to better organize the paper, reduce the a priori judgements, be more descriptive and clarify when they are interpreting the data.

Author Response

Our responses are described in the attached corrections letter.

Reviewer 2 Report

Congratulation for your work!

I have not identified a method for extracting qualitative data and no statistical indicators to validate the research. I suggest prototypical analysis, which is a standard methodology for studying the topic.

This allows to locate the elements of the central representative system through a calculation of frequency and the average rank of the appearance of the terms generated by an inductive word, which allows to locate the central elements. Categorical analysis is a second step, which consists in grouping terms that are semantically close and also in doing different statistical analyzes of these categories.

Author Response

(The authors gave the same response as above.)

Reviewer 3 Report

-what is the reason to consider only the Nova Liga 15 Gaúcha de Futebol Infantil through interviews with twelve coaches and in loco observations?

-keywords can be provided at least 5 where most familiar words in the work

- organization of the paper can be provided in the introduction section without fail as the last paragraph

-The related work section needs to be revised to include more relevant researches. It should highlight the outcome of the other work from literature. and these papers can be cited 'an efficient classification of neonates cry using extreme gradient boosting-assisted grouped support vector network'                                                                                                           Examining the relationship between sport spectator motivation, involvement, and loyalty: a structural model in the context of Australian Rules football

-The research data were collected and finalized in the second half of 2018. Why can't you collect the recent data 2022 or 2021

-please check the rows from 169 to 186. given information is not clear. row 193 also . what is the meaning of ..... many places it used. why?

- role of the family is very important to maintain players health. explore more about it

-food habits also very essential. please explain it as one parameter

-The paper need proper proof reading and grammar editing.

Author Response

(The authors gave the same response as above.)

Round 2

Reviewer 1 Report

The authors did a great job in improving the clarity of the paper, specially at the methodological level. This is a good example for the utility of the review process for authors.

However there are still two issues that the authors did't address in a convincingly way (my previous points 6 and 7).

1. Given that the authors classify some categories as positive and other as negative, the rationale for this classification should be previously presented. This means that "the literature addresses on the subject, especially in research related to youth positive development" should be presented before the methods section to explain the classification offered by the authors to coaches reports.

2. The results, qualitative as they are are, should not be a deposit of sentences from coaches. I accept authors position that the discussion can be separate, but in any way the results must be processed and organized (not interpreted) according to the theoretical rationale of the study. In this way every sentence from coaches should be linked to such processing and organization, for the reader to understand why these sentences were selected and not others. Also if the results were not to be processed and organized, authors would simply place the entire transcription of the interviews, which is something the authors surely do not agree. The authors need to clarify their own way of structuring the results by commenting specifically what each of those sentences is informing about.

Author Response

Response to Reviewer 1 Comments

Point 1. Given that the authors classify some categories as positive and other as negative, the rationale for this classification should be previously presented. This means that "the literature addresses on the subject, especially in research related to youth positive development" should be presented before the methods section to explain the classification offered by the authors to coaches reports.

Response 1. In the last paragraph of the methods section, we added a description of the classification process of the data obtained into positive and negative elements that were based on the opinions of the coaches and supported by the literature cited in the corrected version of the manuscript

Point 2. The results, qualitative as they are, should not be a deposit of sentences from coaches. I accept authors position that the discussion can be separate, but in any way the results must be processed and organized (not interpreted) according to the theoretical rationale of the study. In this way every sentence from coaches should be linked to such processing and organization, for the reader to understand why these sentences were selected and not others. Also if the results were not to be processed and organized, authors would simply place the entire transcription of the interviews, which is something the authors surely do not agree. The authors need to clarify their own way of structuring the results by commenting specifically what each of those sentences is informing about.

Response 2. We returned to the data and the analysis process carried out through the note presented by the reviewer, which did not allow the reorganization of our study data. From this organization of the data, the following categories emerged: 1) “The positive aspects of the family participation”, highlighting the importance of the family’s role, and the importance of the family’s presence; 2) “The negative aspects of the family participation”, with emphasis on the impacts of the family´s presence; and 3) “The Contextual influences on negative and positive family participation”. When we carried out this reorganization, some of the reports presented were removed from the text because we understood that they did not reflect this new categorization in such a faithful and profound way. In other cases, we chose to keep all reports, as we understand their relationship with the proposed categorization. In advance, we thank you for presenting this observation that helped us to better organize the presentation of the results.

Reviewer 2 Report

Dear authors,

The work put into this article was hard, congratulations.

On the other hand, the qualitative interpretation must be supported by the quantitative one, regardless of the technique chosen to highlight your variables.

The conclusions must be verified.

In this case, in qualitative research, based on a small number of cases and on a data analysis methodology that is not extremely well set up chances to make an error are quite large. Miles and Huberman (1995:263) suggest 13 possibilities of verification, oriented towards four objectives:

1) Checking the quality of the data can be done by:

a) Checking the representativeness of the cases;

b) Checking the effects the researcher had on the data (reactivity);

c) Triangulation;

d) Data evaluation – certain data or informants are of better quality;

2) Verification of cases that do not comply with the regulations:

a) Verification of exceptions;

b) Use of extreme cases or situations;

c) Investigating surprising situations;

d) Search for contrary evidence;

3) Testing the explanations offered:

a) Constructing sentences like if... then... and checking them;

b) Elimination of apparent relationships;

c) Replication of results on new data;

d) Examining and eliminating alternative explanations;

4) Obtaining feedback from informants.

I think that the research methodology must be improved.

Author Response

Point 1. [...] On the other hand, the qualitative interpretation must be supported by the quantitative one, regardless of the technique chosen to highlight your variables. The conclusions must be verified. In this case, in qualitative research, based on a small number of cases and on a data analysis methodology that is not extremely well set up chances to make an error are quite large. Miles and Huberman (1995:263) suggest 13 possibilities of verification, oriented towards four objectives:

1) Checking the quality of the data can be done by:

  1. a) Checking the representativeness of the cases;
  2. b) Checking the effects the researcher had on the data (reactivity);
  3. c) Triangulation;
  4. d) Data evaluation – certain data or informants are of better quality;

2) Verification of cases that do not comply with the regulations:

  1. a) Verification of exceptions;
  2. b) Use of extreme cases or situations;
  3. c) Investigating surprising situations;
  4. d) Search for contrary evidence;

3) Testing the explanations offered:

  1. a) Constructing sentences like if... then... and checking them;
  2. b) Elimination of apparent relationships;
  3. c) Replication of results on new data;
  4. d) Examining and eliminating alternative explanations;

4) Obtaining feedback from informants.

I think that the research methodology must be improved.

Response 1: This study is justified as a qualitative approach, with a naturalist position and an interpretative perspective of human experience, thus, aiming to establish an epistemological coherence in all its stages. We did not choose to carry out quantitative analyses. We understood, however, the meaning of the methodological improvements requested by the reviewer and for that, we added greater detail on the qualitative choices of the analytical process. We also added two current references (2023), both published in Sustainability on studies that use qualitative processes similar to ours for the collection, analysis, and reliability of the interpretations made. We understand that this change is opportune to add greater credibility and security to readers.

Reviewer 3 Report

in the related work section, this one can be cited ' An efficient classification of neonates crv using extreme gradient boosting assisted grouped-support-vector network

Author Response

Response to Reviewer 3 Comments

Point 1. In the related work section, this one can be cited ' An efficient classification of neonates crv using extreme gradient boosting assisted grouped-support-vector network

Response 1: We had difficulty understanding the reviewer's indication (whether the study should be included in the references or elsewhere). Nor did we know how the suggested research would be included, as we could not establish relationships due to the discrepancy in theme, content, and format between the two surveys. Therefore, we did not make the change, but we remain open to further clarification from the reviewer on this point.

Round 3

Reviewer 1 Report

I congratulate the authors for the good work.

Author Response

Response to Reviewer 1 Comments

Point 1: I congratulate the authors for the good work.

Response 1: We thank you for your help in qualifying our text.

Reviewer 2 Report

Dear authors,

Fidelity: At a time interval, the application of the research tool is repeated on the same sample, subsequently the results obtained being compared with those obtained at the first application, by means of the correlation coefficient. Thus, how much the higher it is, the more accurate the measuring tool is. It is generally considered that a coefficient of correlation greater than 0.70 between responses to the first and second application indicates high fidelity of the instrument.

The problem regarding the assurance of validity arises both in relation to measurement procedures and in relation to qualitative research, only that in the case of measurement it mostly refers to the evaluation of research instruments and here, measurement is missing. If validity can be easily explained, fidelity must be calculated quantitatively and mandatory in a qualitative research.

I suggest calculating the correlation index to check the instrument's fidelity.

Author Response

Response to Reviewer 2 Comments

Point 1: 

Fidelity: At a time interval, the application of the research tool is repeated on the same sample, subsequently the results obtained being compared with those obtained at the first application, by means of the correlation coefficient. Thus, how much the higher it is, the more accurate the measuring tool is. It is generally considered that a coefficient of correlation greater than 0.70 between responses to the first and second application indicates high fidelity of the instrument.

The problem regarding the assurance of validity arises both in relation to measurement procedures and in relation to qualitative research, only that in the case of measurement it mostly refers to the evaluation of research instruments and here, measurement is missing. If validity can be easily explained, fidelity must be calculated quantitatively and mandatory in a qualitative research.

I suggest calculating the correlation index to check the instrument's fidelity.

Response 1: 

We appreciate the reviewer for the contributions and agree with the need for a judicious and careful organization regarding the methodological structure of scientific research. We also agree on the importance of instruments, data, and quantitative analysis in specific studies, as well as the mix of qualitative elements in research of a mixed nature. However, we also understand qualitative studies as fundamental, which present epistemological coherence, in addition to a careful and judicious process of methodological construction. It is in this context that our study is located and, therefore, we seek to organize all its stages from a qualitative perspective.

Likewise, we base the entire process on literature that is equally focused on a qualitative perspective. In this sense, it would not be adequate and coherent to use quantitative inferences to ensure the fidelity of the analysis carried out, which we understand to be contemplated by the qualitative process. We also seek to ensure coherence and reliability with the guidelines established by Sustainability and, therefore, we have included in the references two recent studies published by the journal whose structure is exclusively qualitative. Are they:

  • Raksmey Sann; Pei-Chun Lai; Chi-Ting Chen. Crisis Adaptation in a Thai Community-Based Tourism Setting during the COVID-19 Pandemic: A Qualitative Phenomenological Approach. Sustainability 2023, 15, 340. https://doi.org/10.3390/su15010340
  • Angela C. Dufour; Fiona E. Pelly; Judith Tweedie; Hattie Wright. Perceptions of the Impact of COVID-19 Countermeasures on Safe Foodservice Provision at International Sporting Competitions: A Qualitative Study. Sustainability 2023, 15, 576. https://doi.org/10.3390/su15010576